Cirsilineol improves anesthesia/surgery-induced postoperative cognitive dysfunction through attenuating oxidative stress and modulating microglia M1/M2 polarization

Du Junli
Chen Chao
Chen Jie 72300100228@shsmu.edu.cn
Department of Anesthesiology, The Ninth People’s Hospital, Shanghai Jiao Tong University School of Medicine , Shanghai , China
Guan Fanglin
Electronic publication date: 2024 Nov 15
Publication date: 2024
Volume: 12
Electronic Location ID: e18507
Received 2024 Aug 21; Accepted 2024 Oct 21
Copyright: © 2024 Du et al.
Copyright year: 2024
Copyright holder: Du et al.
License: This is an open access article distributed under the terms of the Creative Commons Attribution License, which permits unrestricted use, distribution, reproduction and adaptation in any medium and for any purpose provided that it is properly attributed. For attribution, the original author(s), title, publication source (PeerJ) and either DOI or URL of the article must be cited.
License URL: https://creativecommons.org/licenses/by/4.0/

Keywords: Cirsilineol, Oxidative stress, JAK/STAT pathway, Postoperative cognitive dysfunction, Microglia polarization

Funding: The authors received no funding for this work.

==============================
Background

Cirsilineol is a trimethoxy and dihydroxy flavonoid isolated from plant species such as Artemisia vestita and has a variety of pharmacological properties. This study analyzed whether cirsilineol could prevent postoperative cognitive dysfunction (POCD).

Methods

A POCD mouse model induced by anesthesia/surgery induction and a cell model established with hydrogen peroxide (H2O2)-induced microglia BV-2 were employed to explore the efficacy of cirsilineol on POCD. The cognition function of the mice were assessed by carrying out behavioral tests (Morris water maze test and Y-maze test). We assessed the activation and polarization status of microglia using immunofluorescence analysis and detected the expression levels of CD86 and CD206 using the quantitative PCR (qPCR). Subsequently, cell viability was determined by CCK-8 assay and apoptosis was assessed using Calcein-AM/PI staining. Meanwhile, superoxide dismutase (SOD) and malondialdehyde (MDA) levels in plasma and cell culture medium were detected using chemiluminescence. Finally, the phosphorylation levels of JAK/STAT signaling pathway-related proteins were analyzed by Western blot.

Results

Cirsilineol reduced the escape latency and times of crossing island and increased spontaneous alternation (SA) rate, restoring the cognitive dysfunctions of POCD-modeled mice. Meanwhile, POCD elevated CD86 expression and malondialdehyde content and lowered the level of SOD; however, cirsilineol promoted CD206 expression and generation of SOD and inhibited malondialdehyde production. In H2O2-induced microglia BV-2, cirsilineol treatment increased SOD content and suppressed the generation of reactive oxygen species (ROS) and malondialdehyde, modulating microglia M1/M2 polarization and JAK/STAT pathway.

Conclusion

Cirsilineol prevented against POCD by attenuating oxidative stress and modulating microglia M1/M2 polarization, providing novel insights for the management of POCD.

Introduction

A study estimated that 50% of the elderly population receive at least one surgery in their lifetime (Kotekar, Shenkar & Nagaraj, 2018). Neurocognitive disorders are common complications of surgery. Postoperative cognitive dysfunction (POCD) refers to a declined cognitive performance that occurs to 10–25% patients 3 to 6 months after taking surgery (Feinkohl et al., 2023; Borchers et al., 2021). Recent discoveries emphasize an underlying relationship between delirium and POCD for patients whose brains may be vulnerable to cognitive decline under a stress condition from surgery and anesthesia (Vacas, Cole & Cannesson, 2021). POCD is characterized as dysfunctions of memory, mental capacity, language capacity, or other cerebral function, which can remain for months or even years after anesthesia and surgery and even cause a permanent disorder in some cases (Lin et al., 2020; Alalawi & Yasmeen, 2018).

A study showed that neuroinflammation may be an underlying pathophysiology of POCD (van Zuylen et al., 2021). Therefore, therapeutic modulation on the immune processes that initiate and maintain neuroinflammation has been widely studied. At present, the development of anti-neuroinflammatory drugs is hindered by a lack of reliable biomarkers (Granger & Barnett, 2021). The efficacy of natural products on treating POCD has been analyzed by some preliminary explorations and randomized controlled trials. For instance, Valeriana officinalis root extract given to POCD patients every 12 h for a total of 8 weeks could improve the cognitive state of the patients (Hassani et al., 2015). Cistanche exerts antioxidant and anti-inflammatory effects on a sevoflurane-induced POCD model in aged rat (Peng et al., 2020). Further, injection of Shenmai and Shenfu, two Chinese medicine formulations containing ginseng, could facilitate conscious recovery and prevent postoperative cognitive decline, as shown by minimized surgery-induced inflammation and reduced perioperative stress responses (Zhang et al., 2018).

Flavonoids are plant-derived natural products containing potent antioxidant and biological properties that are conducive to human health (Xu et al., 2020; Rodríguez-Arce & Saldías, 2021). Cirsilineol is a trimethoxy and dihydroxy flavonoid isolated from plant species such as Artemisia vestita and has anti-oxidant and anti-inflammatory potentials (Yoon et al., 2011; Yin et al., 2008). A preliminary animal study showed that cirsilineol alleviates PM2.5-induced lung injury in mice through modulating mammalian target of rapamycin (mTOR) and toll-like receptor (TLR) 2, 4/MyD88 pathways (Kim, Kim & Bae, 2022). Cirsilineol exerts a gastroprotective effect on hydrochloric acid/ethanol-induced gastric ulcer rat model (Gong et al., 2021). Moreover, cirsilineol plays a protective role against ovalbumin-induced allergic rhinitis in mice through inhibiting inflammation and oxidative stress (Gong et al., 2021). Currently, the effects of cirsilineol on cerebral disorders remained to be comprehensively analyzed, to bridge the gap, this study aimed to explore the efficacy of cirsilineol on POCD using an anesthesia/surgery-induced model in aged mice. We have revealed for the first time the protective effect of cirsilineol in POCD and provided new insights that it may achieve this effect through the modulation of the JAK/STAT signaling pathway, providing a new direction for the future development of therapeutic agents for POCD.

Materials and Methods

Ethics

All the animal experiments in this study were reviewed and approved by The Ninth People’s Hospital, Shanghai Jiao Tong University School of Medicine Animal Experimental Ethics Committee (SH9H-2024-A1018-SB) and performed according to the guidelines of China Council on Animal Care and Protocol and the ARRIVE guidelines. Effort has been made to minimize the suffering to the animals.

POCD mouse model construction

Aged male C57BL/6J mice (18 months old, body weight: 25–30 g) were purchased from the Shanghai Experimental Animal Research Center (Shanghai, China) and reared in groups (22–25 °C) under a 12-h circadian cycle and provided with free access to water and food.

All the mice were randomly grouped into Sham, Anesthesia (A)+Surgery (S)+ phosphate buffered saline (PBS) (A+S+PBS) (anesthesia and surgical management, and PBS was given), and A+S+CSL (anesthesia and surgical management, and received cirsilineol treatment) (n = 6 for each group) and unbiased double-blinded experiments were performed during the daytime. Specifically, the mice in Sham group only received the anesthesia and analgesia without any other surgery procedure; the mice in the A+S+PBS group were given equal volume of PBS solution before the anesthesia and surgery; the mice in the A+S+CSL group were given cirsilineol via intraperitoneal injection at 10 mg/kg for seven continuous days before the anesthesia and surgery. In this study, 10% dimethyl sulfoxide (DMSO, ST038; Beyotime, Shanghai, China), 40% PEG300 (HY-Y0873; MedChemExpress, Monmouth Junction, NJ) and 50% saline were used to dissolve Cirsilineol (HY-119347; MedChemExpress, Monmouth Junction, NJ, USA) as needed.

The POCD mouse model was established by referring to a previous study (Liu et al., 2021). Following 1-week acclimation, the mice were given 3% isoflurane (C153359; Aladdin, Shanghai, China) for anesthesia induction and 1.5% isoflurane for maintenance. An incision was made on the lateral side of the left tibia to expose the bone. An intramedullary fixation pin (0.3 mm) was implanted to the spinal canal from the hole drilled at the trochanter of the tibia. Next, osteotomy was performed at the middle and distal tibia, and the incision was closed with 5-0 Vicryl sutures. The body temperature of mice was monitored and maintained at 37 ± 0.5 °C using a heating pad during the operation. The mice were put back to the home cages after they had recovered from anesthesia and surgery. A total of 2% lidocaine (Shiyao Yinhu Pharmaceutical Co., Ltd., Sichuan Sheng, China) was locally applied for the treatment of postoperative pain twice per day for three successive days.

Behavioral tests

The behavioral tests in the current study included Morris water maze test and Y-maze test. For the Morris water maze test, a circular pool (depth: 50 cm, diameter: 120 cm) pre-filled with water (depth: 38 cm) was painted with tempera for spatial cues. A platform (diameter: 10 cm) was placed 1 cm below the water (water temperature was set at room temperature of 22 °C). In the training phase, the mice were instructed to familiarize themselves with water and swimming for 3 days (five training sessions per day) and had to locate the platform within 1 min. Then the mice were allowed to rest on the platform for 15 s. Those who failed to find the platform within 1 min were guided to the platform and also allowed to rest on the platform for 15 s. For the testing phase, the platform was removed and the mice were allowed to swim freely in the pool for 90 s in a total of 4 days. The relevant trajectory was recorded and the escape latency (s) was calculated by the experimenters blinded to the allocations of the animals (Wei et al., 2021).

For the Y-maze test, a Y-maze apparatus (40 cm × 5 cm × 10 cm) with two equal arms and one different arm separated by 120° was applied (Kraeuter, Guest & Sarnyai, 2019). In the first part of the test, one of the arms were blocked from the opening ones with an opaque board with the same texture of the apparatus. The mice were put at the end of the arm and allowed to explore the unblocked arms for 8 min. After 1 h, the blocking board was removed to allow the mice to explore for another 8 min as the second part of the test. The spontaneous alternation (SA) rate (%) was calculated by the experimenters blinded to the allocations of animals using the following formula: SA (%) = [number of alternations/(total number of arm entries -2)] × 100%.

At the end of the experiments, all the mice were sacrificed via inhalation of CO2 overdose, and the samples were collected for subsequent use.

Cell culture and intervention

Dulbecco’s modified Eagle’s medium (DMEM, 11966-025; Gibco, Grand Island, NY, USA) added with 10% fetal calf serum (10099-141C; Gibco, Waltham, MA, USA) and 1% penicillin-streptomycin (15070-063; Gibco, Waltham, MA, USA) was used to culture the murine microglia cell line BV-2 (CL-0493; Procell, Wuhan, China) at 37 °C with 5% CO2. Next, the cells were grouped as follows: (1) Control (Con): BV-2 cells were normally cultured without intervention; (2) H2O2: BV-2 cells were exposed to 100 μM H2O2 for 24 h; (3) H2O2 + CSL: BV-2 cells were pre-treated with cirsilineol (5, 10, 20, 40 and 80 μM) for 24 h and exposed to 100 μM H2O2 for another 24 h.

Cell viability test

The Cell Counting Kit-8 (CCK-8) assay kit (C0037; Beyotime, Beijing, China) was applied for the cell viability test. BV-2 cells of each group were seeded into 96-well plates at the density of 5 × 103 cells per well and supplemented with 10 μL CCK-8 solution for 3-h culture at 37 °C. A microplate reader (Thermo Fisher Scientific, Waltham, MA, USA) was used for reading at 450 nm.

Measuring the ROS content

The cellular ROS content in BV-2 cells of each group was quantified with a commercial assay kit (E-BC-K138-F; Elabscience, Hoston, TX, USA). In detail, the DCFH-DA working solution was prepared using 10 μM serum-free medium and added to BV-2 cells for incubation for 30 min at 37 °C. Then, the cells were re-suspended in serum-free medium, and the fluorescence was read at the excitation and emission wavelengths of 500, 525 nm, respectively.

Calcein-AM/propidium iodide (PI) staining

BV-2 cells with different stimulations were collected and centrifuged at 1,000 g for 5 min at room temperature to collect cell pellet, which were rinsed in PBS. Next, BV-2 cells were dyed with 2 μM Calcein-AM and 4.5 μM PI provided by the assay kit (C2015M; Beyotime) at 37 °C for 30 min. A confocal laser scanning microscope (LSM800; Zeiss, Oberkosen, Germany) was applied to take the relevant imgaes (Qiu et al., 2019).

Immunofluorescence

BV-2 cells with different treatment were washed with pre-warmed PBS and fixed with 4% paraformaldehyde (P0099; Beyotime, Beijing, China) for 10 min and then permeabilized with 0.2% Triton X-100 (ST1722; Beyotime, Beijing, China) for another 10 min at room temperature. For the brain tissues, the mice were sacrificed and perfused with saline and 4% paraformaldehyde in 100 mM PBS. The brain tissue was collected, fixed in 4% paraformaldehyde overnight and immersed in 30% sucrose for 48 h. Then the coronal sections were resected into a thickness of 20 μm and permeabilized in 0.1% Triton X-100 for 30 min. Confocal microscopy (LSM800; Zeiss, 25X lens) was performed to observe and record the signals of Iba-1, CD86, and CD206.

Following washing with PBS, the brain tissues and BV-2 cells were incubated with the blocking buffer at 4 °C for 1 h and the primary antibodies against Iba-1 (019-19741; 1:1,000, Wako Fujifilm, Osaka, Japan), CD86 (942-RBM-4-P1; 1:1,000, Thermo Fisher Scientific) and CD206 (CL594-60143; 1:100, Proteintech, Wuhan, China) at 4 °C overnight. Subsequently, Alexa Fluor 488-conjugated goat anti-rabbit IgG secondary antibody (ab150077; 1:1,000, Abcam, Cambridge, UK) was further used to incubate with the sections. The nuclei were finally counterstained with DAPI (C1002; Beyotime) and then the sections were rinsed in PBS three times and observed under a confocal laser microscope. The affiliated software was applied to calculate the relative mean fluorescence intensity (MFI).

Quantification of the oxidative stress indicators

The blood samples were collected by opening the thoracic cavity of the mice and centrifuged at 2,000 rpm for 10 min to harvest the plasm; meanwhile, the culture media of BV-2 cells in each group were harvested and centrifuged to collect the supernatant. The content of SOD and malondialdehyde (MDA) was quantified according to the instructions of the assay kits.

Quantitative PCR (qPCR)

A commercially available RNA isolation kit (RE-03113; Foregene, Beijing, China) was applied to separate total RNA from the brain tissue and cell samples. ChamQ Universal SYBR qPCR Master Mix (Q711-02; Vazyme, Nanjing, China) and Step One Plus Real-Time PCR System (Thermo Fisher Scientific, Waltham, MA, USA) were employed in PCR procedure (Sindhuja, Amuthalakshmi & Nalini, 2022). Primers used in this study were designed by PrimerBank and NCBI and validated with BLAST algorithm. Relative gene expression was calculated with the 2−ΔΔCt method (Livak & Schmittgen, 2001). The sequences used were listed in Table 1.

Table 1 Sequences of primers.

Gene	Forward primer sequence (5′-3′)	Reverse primer sequence (5′-3′)	
Cd86	ACGTATTGGAAGGAGATTACAGCT	TCTGTCAGCGTTACTATCCCGC	
Cd206	GTTCACCTGGAGTGATGGTTCTC	AGGACATGCCAGGGTCACCTTT	
Gapdh	CATCACTGCCACCCAGAAGACTG	ATGCCAGTGAGCTTCCCGTTCAG	

Western blot

Total protein from BV-2 cells of each group was lysed and extracted using the lysis buffer containing phosphate and protease inhibitor (P0013C; Beyotime) and the concentration was quantified using BCA protein assay kit (BL521A; Biosharp, Hefei, China). Subsequently, equivalent volume of protein sample was separated in 12% SDS-PAGE separation gel and transferred to PVDF membrane, which was blocked by 5% bovine serum albumin (BSA). Next, the film was incubated with primary antibodies at 4 °C overnight and then with secondary antibodies at room temperature for 1 h. After rinsing in TBST, the Tanon 5200 Multi Gel Imaging Analysis System (Tanon, Shanghai, China) was applied for the detection, and the results were processed with Gel-Pro Analyzer software 4.0 (Media Cybernetics, Rockville, Maryland, MD, USA). Relevant antibodies used were listed in Table 2.

Table 2 Information of antibodies.

Identifier	Catalog No.	Host species	Molecular weight (kDa)	Dilution ratio	Manufacturer	
Jak1 Antibody	#3332	Rabbit	130	1/1,000	CST	
Phospho-Jak1(Tyr1034/1035) (D7N4Z) Rabbit mAb	#74129	Rabbit	130	1/1,000	CST	
Stat1 Antibody (SM2)	sc-51702	Mouse	91	1/1,000	Santa Cruz Biotechnology	
Recombinant Anti-STAT1 (phospho S727) antibody (EPR3146)	ab109461	Rabbit	91	1/10,000	Abcam	
Recombinant Anti-STAT6 antibody (YE361)	ab32520	Rabbit	94	1/2,000	Abcam	
Recombinant Anti-STAT6 (phospho Y641) antibody (EPR22599-78)	ab263947	Rabbit	94	1/1,000	Abcam	
Anti-GAPDH antibody (6C5)–Loading Control	ab8245	Mouse	36	1/10,000	Abcam	
Goat anti-rabbit IgG-HRP	sc-2004	Goat	/	1/10,000	Santa Cruz Biotechnology	
Goat anti-mouse IgG-HRP	sc-2005	Goat	/	1/10,000	Santa Cruz Biotechnology	

Statistical analysis

Data from at least three independent trials were expressed as mean ± standard deviation. GraphPad Prism 6.0 was applied for statistical analyses. We tested the data for normality to ensure that the data used met the assumption of a normal distribution. Notably, by using Student’s t-test in order to be used for comparison of differences between two groups, while one-way ANOVA was used for comparison of differences between three and more groups. A P-value lower than 0.05 were regarded as statistically significant.

Results

Cirsilineol restored the cognition function of the POCD mice

An anesthesia/surgery-induced mice model was established to investigate the potential effect of cirsilineol on POCD. The trajectories of the mice in each group during the Morris water maze test were displayed in Fig. 1A, and the escape latency, swimming speed and times of crossing island were also recorded. Following the modeling procedure, a longer escape latency on day 1 to day 4 and fewer times of island crossing were observed (Figs. 1B, 1D, P < 0.01) but the swimming speed was not affected (Fig. 1C, P > 0.05). However, the intervention of cirsilineol reduced the escape latency and increased the times of crossing island (Figs. 1B, 1D, P < 0.01) but did not evidently affect the swimming speed (Fig. 1C, P > 0.05). After the modeling procedure, the results of Y-maze test showed that the starting arm (SA) rate of the mice in each group was evidently reduced (Fig. 1E, P < 0.0001), while the administration of cirsilineol promoted the SA rate in the mice (Fig. 1E, P < 0.001). Collectively, these results indicated the potential of cirsilineol on the restoration of the cognition functions in the POCD model mice.

Figure 1 Cirsilineol contributes to the restoration on the cognition of postoperative cognitive dysfunction-model mice.

(A–D) Tracked trails of mice in Morris Water Maze test, along with the quantified escape latency (s), swimming speed (cm/s), and number of island crossing. (E) Quantified spontaneous alternation rate in each group of mice based on Y-maze test. Results were expressed as mean ± standard deviation. ns represents non-significant; P > 0.05; **P < 0.01; ***P < 0.001; ****P < 0.0001.

Cirsilineol repressed the microglial activation and oxidative stress in the POCD mice

To determine the activation and polarization state of microglia in POCD and to explore the role of cirsilineol in its regulation, the immunofluorescence assay was used to evaluate the activation of microglia using the Iba-1 antibody and to quantify the expression levels of relevant microglia markers (Cd86 for M1 and Cd206 for M2). As shown by an intense red fluorescence, the modeling resulted in a sharp elevation of Iba-1+ cells, (Figs. 2A, 2B, P < 0.0001), which was consistent with the upregulated expression of Cd86 (Fig. 2C, P < 0.01). The expression level of Cd206 was almost unchanged (Fig. 2D, P > 0.05). The intervention using cirsilineol reduced Iba-1+ cells (Figs. 2A, 2B, P < 0.0001), downregulated Cd86 expression but upregulated Cd206 expression in the POCD mice (Figs. 2C, 2D, P < 0.01).

Figure 2 Cirsilineol repressed the microglial activation and oxidative stress in POCD-modeled mice.

(A and B) Iba-1+ cells in each group of mice based on the quantification on the fluorescence intensity in immunofluorescence assay. (C and D) The calculated levels of microglia polarization indicators Cd86 (M1) and Cd206 (M2). (E and F) The gauged levels of SOD and MDA in plasm of mice in each group. Results were expressed as mean ± standard deviation. ns represents non-significant; P > 0.05; *P < 0.05; **P < 0.01; ***P < 0.001; ****P < 0.0001.

The extent of oxidative stress in vivo was tested based on the quantified results on the levels of anti-oxidant indicator SOD and the peroxidation product MDA. It was found that the POCD modeling lowered the level of SOD but increased that of MDA (Figs. 2E, 2F, P < 0.01), whereas cirsilineol suppressed the generation of MDA but increased that of SOD in the POCD mice (Figs. 2E, 2F, P < 0.05). These results suggested the potentials of using cirsilineol to inhibit microglial activation and oxidative stress in POCD.

Cirsilineol promoted the survival of H2O2-treated microglia but suppressed ROS generation

Then H2O2-treated microglia were used to further examine the anti-inflammatory effects of Cirsilineol. The results confirmed the inhibitory effects of H2O2 on the viability of BV-2 cells (Fig. 3A, P < 0.0001) and its promoting effects on the production of ROS in BV-2 cells (Fig. 3B, P < 0.0001). However, cirsilineol, restored the viability of BV-2 cells and suppressed ROS production (Figs. 3A, 3B, P < 0.0001). Considering the efficiency of restoring the cell viability, 20 μM cirsilineol was applied in subsequent assays. The results from Calcein-AM/PI staining revealed a pro-apoptosis effect of H2O2 in BV-2 cells, as shown by increased fluorescence (Fig. 3C), which, however, was reduced by the treatment of cirsilineol (Fig. 3C). Collectively, these results indicated the pro-survival and anti-ROS potential of cirsilineol in H2O2-treated microglia.

Figure 3 Cirsilineol promoted the survival yet restrained ROS generation in H2O2-treated microglia.

(A) Quantified viability of BV-2 cells of each group based on CCK-8 assay. (B) The calculated ROS content of BV-2 cells of each group. (C) Representative images of Calcein-AM/PI staining of BV-2 cells in each group. ns represents non-significant; P > 0.05; ****P < 0.0001.

Cirsilineol suppressed the microglial M1 polarization and oxidative stress in H2O2-treated microglia

Next, we based an in vitro cellular model to explore the effects of cirsilinel on microglia polarization and oxidative stress. Similar to the in vivo results, the exposure of microglia BV-2 to H2O2 has led to an increase in the MFI of CD86 (Figs. 4A, 4C, P < 0.0001), while that of CD206 was not affected (Figs. 4B, 4D, P > 0.05). A reduced MFI of CD86 but an enhanced MFI of CD206 in H2O2-treated microglia were observed following the intervention of cirsilineol (Figs. 4A–4D, P < 0.001). Also, H2O2 exposure evidently reduced the content of SOD and significantly increased the level of MDA in the microglia (Figs. 4E, 4F, P < 0.01), while such a trend was reversed by the treatment of cirsilineol (Figs. 4E, 4F, P < 0.01).

Figure 4 Cirsilineol repressed the microglial M1 polarization and oxidative stress in H2O2-treated microglia.

(A–D) Quantified CD86 MFI (A, C) and CD206 MFI (B, D) in each group based on immunofluorescence assay. (E, F) The gauged levels of SOD and MDA in the supernatant of cell culture medium in each group. Results were expressed as mean ± standard deviation. **P < 0.01; ***P < 0.001; ****P < 0.0001.

Cirsilineol modulated JAK/STAT pathway in H2O2-treated microglia

To explore the potential neuroprotective effects of cirsilinelo on POCD, we probed the mechanism of its regulation of the JAK/STAT signaling pathway. Here, we calculated the phosphorylation degrees of relevant proteins (JAK1, STAT1, and STAT6) of JAK/STAT pathway in H2O2-treated microglia. The phosphorylation levels of JAK1 and STAT1 were increased following H2O2 exposure (Figs. 5A, 5B, P < 0.01), whereas the level STAT6 was not affected significantly (Figs. 5A, 5B, P > 0.05). After the treatment of cirsilineol, the phosphorylation levels of JAK1 and STAT1 were reduced but that of STAT6 was increased evidently (Figs. 5A, 5B, P < 0.01). It could be concluded that cirsilineol modulated JAK/STAT pathway in H2O2-exposed microglia.

Figure 5 Cirsilineol modulated JAK/STAT pathway in H2O2-treated microglia.

(A and B) Quantification on the phosphorylation of JAK/STAT pathway-relevant proteins in BV-2 cells of each group. Results were expressed as mean ± standard deviation. **P < 0.01; ***P < 0.001; ****P < 0.0001.

Discussion

The current study preliminarily investigated the effects of cirsilineol on anesthesia/surgery-induced POCD using both animal and cell models and explored the molecular mechanisms related to oxidative stress and microglia M1/M2 polarization. Cirsilineol attenuated the oxidative stress and modulated the microglia M2 polarization in both POCD-modeled mice in vivo and H2O2-exposed microglia in vitro. Collectively, the present findings demonstrated the potential of cirsilineol in POCD-model mice in vivo and H2O2-exposed microglia.

The mechanisms of POCD have been proposed to be related to neuroinflammation and oxidative stress (Lin et al., 2020). Animal and human studies showed that neuroinflammation triggered by surgery could lead to POCD (Han et al., 2022). Microglia as a part of the initial response to surgical trauma in the cerebral nervous system can generate massive proinflammatory mediators (Laprell et al., 2021). Microglia can be polarized toward M1-like and M2-like activation states in response to the disturbance of brain homeostasis such as injury or disease, showing distinct effects on tissue repair and neurodegeneration (Shao et al., 2022). Various inflammatory neurodegenerative diseases will occur in the presence of M1 microglia (marked by CD86), which can produce the pro-inflammatory cytokines, ROS and nitric oxide and further contribute to the dysfunction of neural network within the central nervous system (Kwon & Koh, 2020; Nakagawa & Chiba, 2014). In contrast, M2 microglia (marked by CD206) can release the anti-inflammatory mediators and trigger the anti-inflammatory mechanisms and neuroprotectivity (Guo, Wang & Yin, 2022). Previous studies explored the involvement of microglia M1/M2 polarization under the conditions of neuroinflammatory injury, intracerebral hemorrhage (ICH), experimental neuropathic pain, and POCD (Yang et al., 2017; He et al., 2020; Wang et al., 2023). As a class of dietary polyphenols derived from plants, the neuroprotective and pro-cognitive effects of flavonoids have been widely explored (Davinelli et al., 2023). Experimental data confirmed that dietary components containing flavonoids can scavenge pro-inflammatory agents and neurotoxic species generated in the brain as a result of aging (Davinelli et al., 2016). When examining the mechanisms underlying the effects of flavonoids on the brain, some studies suggested that flavonoids could indirectly improve brain health through affecting the microbiota-gut-brain axis, and that flavonoids can exert an anti-amyloidogenic effect and reduce a loss of dopaminergic neurons in the brain (Wang et al., 2023; Putteeraj et al., 2018). Furthermore, flavonoids could protect the central nervous system by suppressing neuroinflammation and protect neurons against stress-induced injury, thereby improving the cognitive function of the brain (Spencer, 2008). Some studied suggested that flavonoids rutin, luteoloside and apigenin are effective on treating POCD (Ji et al., 2024; Zhang et al., 2023; Chen et al., 2017). Cirsilineol is one of the flavonoids in Artemisia vestita with both anti-inflammatory and immunosuppressive effects (Yin et al., 2008; Moufid & Eddouks, 2012; Dogra et al., 2023). The results of this study supported that cirsilineol could protect against anesthesia/surgery-induced cognitive dysfunction in aged mice through modulating oxidative stress and microglia M1/M2 polarization.

Signaling pathways including TLR4/nuclear factor-kappa B (NF-κB) pathway and interferon-gamma signaling play critical roles in realizing the effects of cirsilineol (Ai et al., 2021; Sun et al., 2010). Here, we focused on the modulatory effect of cirsilineol on JAK/STAT pathway, which is stimulated by cytokines and is present and active in the central nervous system (Nicolas et al., 2013). Overactivation of JAK/STAT signaling is related to a variety of neurocomplications, including neuroinflammation, apoptosis, and oxidative stress, and microglia polarization (Kumar, Mehan & Narula, 2023; Li et al., 2022). The JAK/STAT signaling pathway consists of three main components: cellular receptors, JAK proteins, and STAT proteins. The JAK family is comprised of a group of non-transmembrane tyrosine kinases, primarily including four members: JAK1, JAK2, JAK3, and TYK2, which have molecular weights between 120 and 140 kDa. In mammals, there are seven members of the STAT family: STAT1, STAT2, STAT3, STAT4, STAT5A, STAT5B, and STAT6 (Banerjee et al., 2017; Sun et al., 2023). Each STAT family member can be activated by various cytokines alongside their associated JAKs (O’Shea et al., 2015). Initially, cytokines attach to their specific transmembrane receptors, leading to dimerization followed by the activation of JAK kinases that couple with and phosphorylate the receptors. Subsequently, the tyrosine residues within the catalytic domain of the receptor are phosphorylated, creating a docking site for STAT proteins that possess SH2 domains. These STAT proteins are recruited to this docking site, where they undergo phosphorylation, resulting in the formation of either homodimers or heterodimers. Ultimately, the dimerized STAT proteins detach from the receptors and move into the nucleus, where they interact with DNA-binding sites to regulate gene transcription (Muller, 2019; Hu et al., 2023). Past research demonstrated that class I histone deacetylases can enhance neuroinflammation in the hippocampus via JAK/STAT pathway to cause POCD (Yang et al., 2020). Furthermore, existing studies also confirmed the modulatory effects of flavonoids on JAK/STAT pathway under diverse biological conditions (Han et al., 2023; Wang et al., 2022; Li et al., 2023). In our study, we observed that cirsilinel was able to affect the phosphorylation levels of JAK1, STAT1, and STAT6, which revealed that cirsilineol regulates the JAK/STAT signaling pathway by inhibiting the phosphorylation of JAK1 and STAT1 and activating STAT6, which then exerts its neuroprotective effects.

It is worth noting that there are some limitations to our study. For example, this study used an anesthesia/surgery-induced mouse model of POCD, which, although it was able to mimic some of the features of postoperative cognitive dysfunction, was not able to fully represent the complex pathological process of human POCD. Differences in animal models may limit the clinical translatability of results. Future studies could consider using an animal model that more closely resembles human pathologic features to further validate the effects of cirsilineol. In addition, studies in mice of different ages could be considered to more fully assess its effectiveness in different age groups. In addition, this study focused on the therapeutic efficacy of cirsilineol in the early stages of POCD and did not evaluate its long-term efficacy and potential side effects. Therefore, we will add a long-term follow-up experiment to the further part of the study to observe the effect of cirsilineol in prolonged use and its sustained effect on cognitive function. At the same time, its potential side effects, such as effects on liver and kidney function or other organs, should be evaluated.

Collectively, based on the animal and cell models, we investigated the effects of cirsilineol on anesthesia/surgery-induced POCD. We found that cirsilineol significantly improved cognitive function in POCD mice, inhibited M1 polarization and promoted M2 polarization in microglia. Meanwhile, cirsilineol attenuated oxidative stress, increased superoxide dismutase (SOD) levels, and reduced malondialdehyde (MDA) production. In addition, cirsilineol modulated the JAK/STAT signaling pathway by inhibiting the phosphorylation of JAK1 and STAT1 and activating STAT6. This provides new evidence for cirsilineol as a potential therapeutic agent for POCD.

Supplemental Information

Supplemental Information 1 Author Checklist.

Supplemental Information 2 MIQE checklist.

Supplemental Information 3 WB for Original blot.

Abbreviations

POCD Postoperative cognitive dysfunction

mTOR mammalian target of rapamycin

TLR toll-like receptor

DMSO dimethyl sulfoxide

SA spontaneous alternation

DMEM Dulbecco’s modified Eagle’s medium

Con Control

H2O2 hydrogenase peroxide

CCK-8 Cell Counting Kit-8

ROS reactive oxygen species

PI Propidium Iodide

MFI mean fluorescence intensity

SOD superoxide dismutase

MDA malondialdehyde

BSA bovine serum albumin

ICH intracerebral hemorrhage

NF-κB nuclear factor-kappa B

Additional Information and Declarations

Competing Interests

Author Contributions

Animal Ethics

Data Availability

The authors declare that they have no competing interests.

Junli Du conceived and designed the experiments, analyzed the data, prepared figures and/or tables, authored or reviewed drafts of the article, and approved the final draft.

Chao Chen conceived and designed the experiments, performed the experiments, analyzed the data, authored or reviewed drafts of the article, and approved the final draft.

Jie Chen performed the experiments, analyzed the data, prepared figures and/or tables, and approved the final draft.

The following information was supplied relating to ethical approvals (i.e., approving body and any reference numbers):

The study was approved by The Ninth People’s Hospital, Shanghai Jiao Tong University School of Medicine Animal Experimental Ethics Committee (SH9H-2024-A1018-SB).

The following information was supplied regarding data availability:

The raw data is available in GitHub and Zenodo:

- https://github.com/123Jiechen/Updated-raw-data.git

- 123Jiechen. (2024). 123Jiechen/Updated-raw-data: Updated raw data (v.1.1.1). Zenodo. https://doi.org/10.5281/zenodo.13318532.

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
