# Peer review of "Cirsilineol improves anesthesia/surgery-induced postoperative cognitive dysfunction through attenuating oxidative stress and modulating microglia M1/M2 polarization"

_PeerJ, doi:10.7717/peerj.18507_

## Round 0.1 · original submission · Minor Revisions

I have carefully reviewed the comments from all three reviewers, and I am pleased to inform you that your manuscript has been recommended for minor revision. Please address the remaining minor concerns raised by the reviewers to further improve your manuscript. Once you have made these revisions, kindly resubmit your updated manuscript along with a point-by-point response detailing how you have addressed each comment. We look forward to receiving your revised submission.

·

Basic reporting

The language is clear, the introduction and background show the context well referenced and relevant, and the figures are of high quality and well described.

Experimental design

I have read and reviewed this manuscript with great interest and overall, from this reviewer's perspective, it is an experimental study that has been well-planned and executed. The primary research is original and novel, the research question is well-defined, and the study indicates that the research fills a knowledge gap in the area of ​​oxidative stress and postoperative cognitive dysfunction. Other strengths of the manuscript that I can highlight are the following: the introduction provides sufficient background and includes pertinent references, the research design is adequate, and the methods are repeatable and correctly described. The conclusions are supported by the results obtained.

Validity of the findings

The research provides all the data underlying values ​​derived from the experiment, which are robust, statistically solid, and controlled.

Additional comments

Nevertheless, some points must be addressed to achieve publication quality. I have left some comments hoping that they can help the authors.

My general comments are:
L30: In the abbreviation SPD, please clarify its meaning.

L69: please add a hypothesis.

L80: how was the sample size determined? with what statistical method? Please clarify.

L159: just as was done in this line, I suggest the authors add the country of origin to all the reagents and materials used in the experiment.

L188: Please add the statistical analysis of normality that was used in your study.

L190: specify the variables that were analyzed with each statistical model described.

L300: I suggest the authors discuss the limitations identified in the experiment and write a precise and concrete conclusion of the present study.

·

Basic reporting

This study describes the effect of Cirsilineol on the prevention of postoperative cognitive dysfunction in aged mice. The manuscript would benefit from a review of the English language. The findings are clear and relevant, the cited literature is appropriate, and the structure of the manuscript is adequate.
Details are missing from the methods section of the abstract. It is not specified how oxidative stress was determined, nor which proteins were measured using qPCR or WB. Additionally, the phrase “The oxidative stress, microglia M1/M2 polarization, and survival of H2O2-induced BV-2 cells were determined accordingly” appears incomplete.
In the results section of the abstract, the acronym "SPD" is used but not defined.
In line 62, what does “PM25-induced pulmonary” mean?
In line 81, the meaning of the groups "A+S+PBS" and "A+S+CSL" should be specified.
Please explain the use of 10 mg/kg for 7 days of intraperitoneal Cirsilineol
Please explain, why was the Sham group exposed to anesthesia and analgesia? What type of analgesic was used, and at what dose?
Justify the use of the reported doses of H2O2 and Cirsilineol in the cell culture.
It is necessary to specify the primary and secondary antibodies used, as well as their concentrations, in the materials and methods section for the western blot
Include a discussion about the potential receptors of Cirsilineol in microglia

Experimental design

The experimental design is adequate.
In line 81, the meaning of the groups "A+S+PBS" and "A+S+CSL" should be specified.
Please explain the use of 10 mg/kg for 7 days of intraperitoneal Cirsilineol
Please explain, why was the Sham group exposed to anesthesia and analgesia? What type of analgesic was used, and at what dose?
Justify the use of the reported doses of H2O2 and Cirsilineol in the cell culture.
It is necessary to specify the primary and secondary antibodies used, as well as their concentrations, in the materials and methods section for the western blot

Validity of the findings

The findings are clear and relevant

Reviewer 3 ·

Basic reporting

In this study, Cirsilineol belong to flavonoids and has anti-oxidant and anti-inflammatory potentials. The author demonstrated that the Cirsilineol improves anesthesia/surgery-induced postoperative cognitive dysfunction through attenuating oxidative stress and modulating microglia M1/M2 polarization in POCD model. This study is rich in content and clear in logic, which is of great significance to promote the clinical practice of Cirsilineol, I think this article has high academic value. However, the author still needs to refine the manuscript before publication.
1. Line 30, the spelling error of “SPD”. In the results of abstract, the author should clear the role of the expression CD86, superoxide dismutase (SOD), CD206 and malondialdehyde production in different model briefly.
2. Line 66-69, Please briefly describe the whole experiment procedure, purpose and some important results.
3. Line 81, the groups of “A+S+PBS and A+S+CSL” represented what, Please encode the correct abbreviations, A is anesthesia and S is surgery.
4. Line 196, the Morris water maze test was performed, but the results are not described in detail, such as the trajectory in the A+S+PBS was more complex, what does this result mean.
5. In addition, Line 197, a longer escape latency, swimming speed and reduced starting arm (SA) represented what. Please describe the principle and purpose of the experiment clearly.
6. Line 207, the role of microglia in POCD is what. Are microglia the main cause of POCD. Please introduce the function of microglia in introduction, or clear the experimental purpose of immunofluorescence assay microglia in Line 207. The two types of microglia represented what.
7. Line 207, is the Iba-1 a marker of microglia, but Line 207, the Iba-1 represent an activator of microglia. There is also no description of Iba-1 in the materials and methods. The Cirsilineol reduced Iba-1+ cells indicated what.
8. Please appropriately describe the purpose of the experiment before each result.
9. Line 240, please add the role of JAK/STAT pathways in POCD progression in introduction.
10. What are the limitations of this article and what work is necessary for the next study.

Experimental design

no comment

Validity of the findings

no comment

---

## Round 0.2 · accepted · Accept

Both reviewers who provided feedback have expressed their satisfaction with the revisions and improvements made to the manuscript. They found the study to be novel, impactful, and well-executed. However, please address the minor comment from Reviewer #1 regarding specifying the normality test used (L198).

While we did not receive feedback from Reviewer #2, given the positive responses from the other reviewers and the thorough revisions you have made, we feel confident in proceeding with acceptance of your manuscript.

·

Basic reporting

I thank the authors for the attention given to my observations and comments from the first revision. I can highlight that your manuscript has improved significantly, so I have no problem accepting it for publication in this journal.

The last observation I can make is on L198: Please indicate if the normality test was Shapiro Wilk or another.

Thank you for the invitation to participate as a reviewer of this manuscript, which I found interesting.

Experimental design

All aspects of the experimental design have been satisfactorily covered.

Validity of the findings

It is a novel and high-impact study.

Additional comments

It is an experimental study that has been well-planned and executed.

Reviewer 3 ·

Basic reporting

In this study, Circilinol belongs to the flavonoid class and has the potential for antioxidant and anti-inflammatory properties. The author demonstrates that in the POCD model, Circilinil improves postoperative cognitive dysfunction caused by anesthesia/surgery by reducing oxidative stress and regulating M1/M2 polarization of microglia. This study has rich content and clear logic, which is of great significance for promoting the clinical practice of Circilinelo. I believe that this article has high academic value.

Experimental design

no comment

Validity of the findings

no comment